# A Review on Concrete Composites Modified with Nanoparticles

Ghasan Fahim Huseien

Department of the Built Environment, College of Design and Engineering, National University of Singapore, Singapore 117566, Singapore; bdggfh@nus.edu.sg; Tel.: +65-83057143

**Abstract:** Recently, various nanomaterials have extensively been used to achieve sustainability goals in the construction sector. Thus, this paper presents a state-of-the-art review involving the uses of different nanomaterials for production of high-performance cementitious, geopolymer, and alkali-activated concrete composites. The effects of nanomaterials on the fresh properties, mechanical properties, and durability of diverse nanoparticle-modified concrete composites are analyzed. The past developments, recent trends, environmental impact, sustainability, notable benefits, and demerits of various nanomaterial-based concrete production are emphasized. It is demonstrated that nanomaterials including $SiO_2$, $Al_2O_3$, $TiO_2$, and $Fe_2O_3$, etc., can be used effectively to enhance the microstructures and mechanical characteristics (such as compressive strength, flexural, and splitting tensile strengths) of the modified concrete composites, thus improving their anti-erosion, anti-chloride penetration, and other durability traits. In short, this communication may provide deep insight into the role of diverse nanoparticle inclusion in concrete composites to improve their overall attributes.

**Keywords:** concrete composites; nanoparticles; strength performance; microstructures





## 1. Introduction

Nanotechnology is certainly a key innovation in the construction sector and has led to a step forward in high-performance building materials with endurance [1]. The outstanding scientific breakthroughs in the nanotechnology field have enabled the effective utilization of diverse nanoscale materials with distinctive characteristics, improving the basic properties of traditional construction components, such as concrete, glass, metals, paints, plastics, wood, and various waste materials including coal products, ceramic and glass powders, and rice husk ash. Currently, nanomaterials are utilized by various industries for a broad array of applications including the manufacture of ceramic, glass, and steel, medicine, water treatment, and coating and insulating roofs and windows [2,3]. Briefly, the significant progress made in the area of nanotechnology and nanomaterials has allowed for the integration of various emergent properties of the materials into their basic structure for further improvement. Consequently, novel concepts emerged to address the durability issues of concrete, wherein the future construction sector can potentially be benefitted with substantially lower service and maintenance expenses [4].

The rapid development of nanomaterial technology (synthesis and characterization tools) has enabled researchers to strategize diverse uses for conductive ultra-small nanoparticles (NPs) of different metal oxides, carbon, and metals. The use of NPs is more advantageous compared to their bulk counterparts with larger (micrometer and above) particle sizes [5,6]. It is now possible to make NPs of diverse morphologies (geometry, shapes, and sizes) including spheres, prisms, hexagons, cubes, wires, tubes, fibers, clay, needles, rods, and so on [7–9]. Figure 1 depicts the NP classification scheme according to their types of sources such as organic, inorganic, and carbon-based [10]. These NPs can generally be categorized in two ways according to their composition: organic carbon based (for example polymeric) and inorganic compound based. Organic NPs, including dendrimer, ferritins, liposomes, micelles, and polymers, are usually biodegradable and non-toxic. Conversely, inorganic NPs are mainly comprised of metals such as aluminum (Al), gold (Au), cadmium

(Cd), cobalt (Co), copper (Cu), iron (Fe), silicon (Si), and zinc (Zn) as well as various metal oxides such as $SiO_2$, $Fe_3O_4$, $ZnO_2$, $FeO_2$, and $TiO_2$. Regardless of their origin, all nanomaterials show unique properties not observed in their bulk counterparts [11,12]. Repeated studies have revealed that the levels of $CO_2$ emissions and energy consumption can appreciably be reduced via the nanotechnology approach, thereby lowering the industrial wastes or byproducts such as iron blast furnace slag, fly ash, and nanoparticulates. Therefore, the wise use of these materials can meet the sustainability standards of the construction industries worldwide in relation to the environment, climate, cost, and human living.

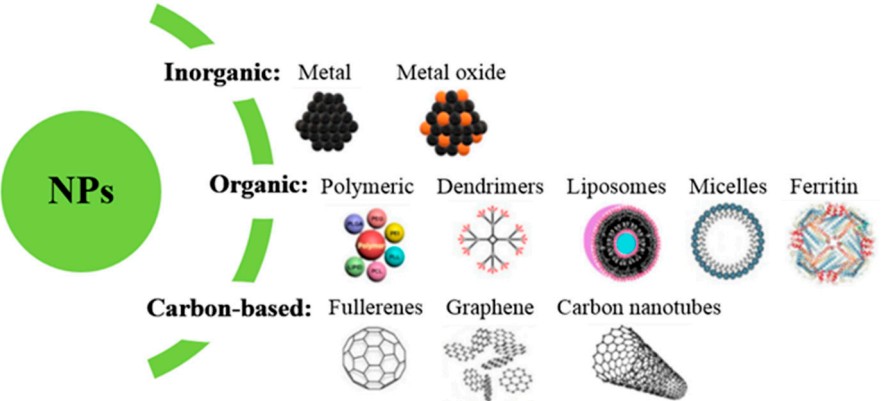

**Figure 1.** Chemical composition-based classifications of NPs [11].

Numerous studies have revealed that different products obtained using nanotechnology approach have varied and unique characteristics that can considerably alter the practices of the current construction sectors, resulting in improved planning and design concepts [1]. For example, the nanostructures of $A1_2O_3$, Ag, Cu, $SiO_2$, and $TiO_2$ can be utilized in diverse ways to coat various construction components (such as floors, roofs, and toilets), thus making them waterproof in addition to the benefits of reduced corrosion and protection against UV radiation [13]. Different studies on green building technology have reported that superior antimicrobial properties can be achieved when paints containing Ag and Cu NPs are used [14,15]. Enhanced electrical, mechanical, and thermal properties were observed in carbon nanotube-reinforced plastics [16]. In addition, various remarkable features shown by carbon nanotubes (CNTs) are highly beneficial for a broad range of applications in electronic devices, optical devices, and material science-based technologies [17]. Advanced nanotechnology has been applied to make basic changes to the properties of various materials. It was found that nanoclay can be widely utilized in the construction fields at a significantly lower cost, thereby driving economy and sustainability in construction. It was shown that montmorillonite clay can be a viable replacement for cement due to its distinctive structure and abundance of silicates [17]. Moreover, using nanostructure-based ceramic and steel composites, an improved resistance to wear and strength can be attained. The addition of NPs to concrete was shown enhance its properties and overall performance. Most research related to nanostructured materials in concretes has displayed that early mechanical strength and general bulk properties can notably be improved.

Considering the immense significance and applied interests of diverse nanomaterial-based sustainable cement or cement-free concretes, the purpose of this paper is to provide a rudimentary overview of the application potential of NP-modified concrete composites in the cement production. Furthermore, it analyzes the uses of various industrial nanomaterials (in the form of wastes/byproducts) as additives to enhance the microstructures of cementitious materials, leading to sustainable advancements in global construction industries. This paper also describes the microscopic mechanisms of the impact of the proposed nanosystems on improving the engineering properties of modified concretes especially

their fresh properties, mechanical properties, durability, and microstructure characteristics. Figure 2 illustrates the overall structure of this review paper.

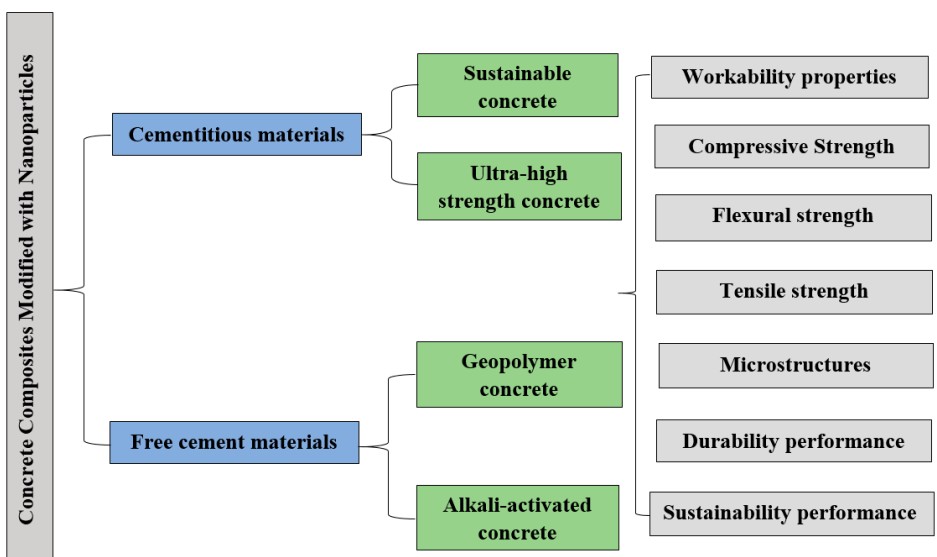

**Figure 2.** Flowchart of the review of various NP-modified concrete composites.

## 2. Nanomaterial-Modified Cement Binder

Throughout the past few decades, various types of nanomaterials have been investigated for the possibility of their inclusion in concrete and further improving the performance of sustainable and ultra-high strength concretes. These nanomaterials are greatly beneficial, because only a small amount of their inclusion can significantly improve the performance of the modified concrete composites. Basically, the nanomaterial-modified cement binders with enhanced morphologies can increase their durability and strength performance [18]. Most the studies in the area of nanomaterial-modified cement, geopolymer, and alkali-activated binders and concretes has focused on the widespread uses of nanosilica [19–22]. Some reports [23–29] have revealed that the cement hydration reaction rate and microstructures are enhanced due to the incorporation of nanosilica, generating high bulk density, better early compressive strength (CS), and improved durability indices.

Table 1 displays the impact of the incorporation of various NP inclusions on ordinary Portland cement (OPC)-based concretes properties, wherein the heat of the hydration mixtures was substantially improved due to the presence of NPs (Figure 3). The observed improvement was ascribed to the larger surface area of NPs, which raised the hydration peak temperature. It was argued that the nanomaterials significantly accelerated the rate of reaction, thereby attaining lower setting times [30,31]. Land and Stephan stated that the increased hydration heat can mainly be attributed to the increased surface area of NP-included concrete [32,33]. Numerous studies have indicated an increase in the hydration heat with an increase in NPs content in the mixes, thereby requiring more water, which could negatively affect the workability and performance of the prepared concrete. Mukharjee and Barai showed that fineness has a considerable impact on the initial and final setting times of different nanomaterial-activated OPC binders [34]. In addition, increasing the fineness of the NPs was found to enhance the cement pastes' reactivity, resulting in greater durability and strength. The accessibility of abundant nucleation centers offered by the NPs was the main reason for the observed improved chemical reaction rate, wherein the rate of hydration reaction was increased and both the initial and final setting times were decreased.

Several reports have suggested that the setting time of the paste can be reduced by increasing the nano-$SiO_2$ content, wherein the profusion of free calcium ions released from the OPC, class C-fly ash (FA) or ground blast furnace slag (GBFS) can produce extra calcium

silicate hydrate (C-S-H) [29,35] and lower the setting time. Additionally, some studies have shown a correlation between increasing NP content and decreasing flowability of the product [36–39]. In comparison to the pure cement paste, pastes incorporated with CNT, $TiO_2$, $Fe_2O_3$, $SiO_2$, $Al_2O_3$, nanoglass, and nanoclay nanomaterials demonstrated substantially lower workability [40,41]. This observation was ascribed to the fineness of NPs that could facilitate greater cohesiveness in the concretes or mortars. Likewise, Hosseini et al. showed that the addition of silica NPs into concrete can notably reduce its workability [42]. The addition of 1.5 and 3% of colloidal nanosilica to 100% recycled coarse aggregate-modified concrete can cause a decrease in the slump flow by 47.1 and 70.59%, respectively. The high reactivity stimulated from the silica NPs with a wider specific surface area was the main reason for the observed decrease in the workability of concrete. Jalal et al. [43,44] identified a lower flowability in concrete due to the incorporation of silica fume (SF) and nano silica (NS). In addition, various properties of the mix such as its consistency, bleeding, and segregation were found to improve in the presence of nanomaterials. According to Joshaghani et al. [45], concrete modified with $TiO_2$, $Al_2O_3$, and $Fe_2O_3$ NPs shows an appreciable decrease in the slump flow. With an increase in NP level, the percentage of water demand in the admixture also increases, thereby accomplishing a slump flow of $650 \pm 25$ mm until it achieved a height of 36% for 5% of $TiO_2$ [45]. Most researchers agreed that the reduced flowability is due to the high reactivity of NPs with larger specific surface area than OPC powder that absorbed more water, thus reducing the concrete's slump [46].

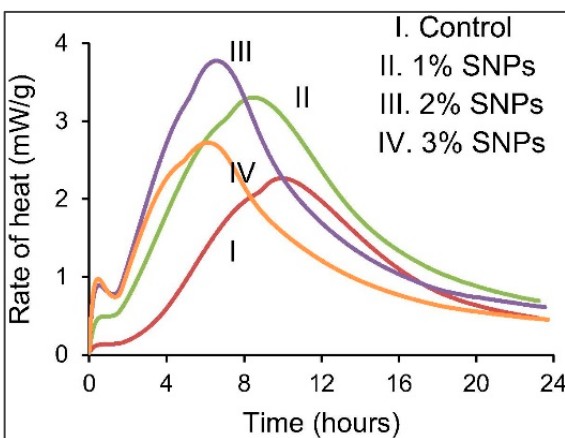

**Figure 3.** Effect of NS contents variation on hydration rate of OPC binders [47].

Over the past two decades, nanomaterials have become more prevalent in cementitious systems because they can enhance the mechanical strength of the products. Many efforts were made to examine how diverse NPs can improve the strength indexes of a nanostructure-modified cement system [48–58]. Table 1 depicts the rise in the early strength of the cement mortars or pastes integrated with diverse oxide-based nanomaterials including alumina, silica, iron oxides, zinc, CNTs, and clay. Zhang et al. [59] examined the impact of white cements with a water to cement ratio (W/C) of 0.5 on the hydration and strength properties when replaced with 1% of silica, $Fe_2O_3$, and NiO NPs followed by three days of curing. The CSs were increased by 23.9, 22.8, 21.9, and 20.5% for the mixes containing 50 nm of $Fe_2O_3$, 15 nm of NiO, 15 nm of $SiO_2$, and 50 nm of $SiO_2$ NPs, respectively. After 28 days of curing, the CSs of the corresponding specimen were increased by 10.2, 9.5, 10.5, and 11.2%, respectively. The observed increase in the CSs of the studied mixes was attributed to the enhanced pozzolanic reactivity, filling impact, extra nucleation sites created by NPs, lower $Ca(OH)_2$ content, and decreased volume of pores.

Said et al. [31] achieved comparable outcomes for a ternary binder incorporated with OPC, FA and silica NPs. The addition of 3 to 6% of nanosilica was found to increase the average CS by 18, 14, and 36% with respective curing ages of 3, 7, and 28 days over the

control samples for all groups. The observed improvement in the strength properties of the binder were attributed to the pozzolanic reaction and NS filling behavior. Kaur et al. [60] investigated the effect of nano-metakaolin on the strength performance of fly ash (FA)-based mortars by replacing FA with 0, 2, 4, 6, 8, and 10% nano-metakaolin. Following three days of ambient curing, the specimen made with 4% of the replacement obtained about 80% of the CS at 28 days of curing age. In comparison to the control group, the increases in the CS after 3, 7, 14, and 28 days of curing were 26.5, 21.4, 21.4, and 22.7%, respectively. Compared to the control sample, the CS values of the specimens made with increasing nano-metakaolin content from 1–10% was decreased by 1–2% for all curing durations.

The effect of NS inclusion into high-level FA-modified cement binder was studied [61]. The results showed that the addition of 3% of nanosilica into the matrix of cement-FA could increase its CS after 7 and 28 days curing by 50% and 10.3%, respectively. Hou et al. [62,63] obtained similar results for mortars made from nanosilica and FA. The observed increase in the CS of the mixtures was mainly due to the presence of fine silica NPs that function as seeds, offering extra nucleation sites to accelerate the hydration process. Furthermore, the pozzolanic trait of nanosilica was evaluated, wherein it was found to react with the lime to generate additional C-S-H gel. This observation was explained using two main reasons. First, following the dissolution of the NS in water, $H_2SiO_4$ was formed that subsequently reacted with the $Ca^{2+}$ thereby producing C-S-H gel. Second, the presence of nanosilica functioned as seeds inside the pores and offered extra nucleation sites, thus improving the pozzolanic characteristics of the product [64].

Figure 4 illustrates the scanning electronic microscopy (SEM) surface morphology of normal cement and cements modified with 3% nanosilica and 3% nano-titanium. The micrographs were comprised of differently sized calcium hydroxide crystallites for the specimens incorporated with nano-$TiO_2$ and nanosilica particles that are smaller in size than those contained in the pure OPC sample. These outcomes verified that NPs are pivotal in relation to filling the large pores and creating additional nucleation sites for calcium hydroxide consumption, controlling the orientations of calcium hydroxide crystals and reducing their size [65]. Control in the calcium hydroxide crystals' (hexagonal-platelet morphology) development could decrease the likelihood of cracks appearing at the interfacial transition zone (ITZ), which is recognized as the zone responsible for managing failure of the mechanical properties (weak zones). The inclusion of NPs could appreciably decelerate the rate of formation of various gel crystals, which is mainly due to the intense reactivity and wide surface area of NPs in the concrete matrix. It was established that the incorporation of diverse NPs in the OPC matrix enabled a deceleration in the calcium hydroxide crystal development rate in the simulated transition regions, generating an extra C-S-H gel and thus functioning as gap/pore-filling agents in the OPC matrix [46].

Generally, the nanomaterials have four key effects in the modified concretes/mortars. Firstly, they function as pore fillers in the concrete matrix, thereby increasing the compactness of mortars. Secondly, these NPs have a high electrostatic force that stimulates hydration in the cement mix and creates additional nucleation sites responsible for subsequent production of additional C-S-H gel clusters. Thirdly, microcrack and pore filling are functions of NPs that enhance the homogeneity and make the network more compact than conventional cement-based concretes devoid of NPs. Fourthly, the high reactivity of NPs indicates a strong chemical reaction with $Ca(OH)_2$ and higher consumption, demanding more water. Hence, they are amassed in the micropores and the ITZ, thereby prompting a refinement process in the microcracks in the initial stages, thus enhancing the microstructures of concrete matrices.

**Table 1.** Effects of NPs on cementitious fresh, strength, and durability performance.

| NP Type | Ref. | Level (%) | Findings |
|---|---|---|---|
| Nanosilica | [47] | 3 | Reduction in flowability by 58%. |
| | [66] | 4 | Reduction in flowability and setting time by 57.2% and 12.5%. |
| | [67] | 3 | Reduction in flowability by 37.5%. |
| | [29] | 3 | Reduction in setting time by 60%. |
| | [30] | 2 | Reduction in setting time by 28%. |
| | [53] | 5 | Improved the strength by 45%, 29.7%, and 10.6% at 3, 7, and 28 days of age, respectively. |
| | [68] | 6 | Improved the strength by 33.5% at 28 days of curing age. |
| | [69] | 6 | Improved the strength by 33% at 28 days of age. |
| | [70] | 4 | Improved the strength by 58 and 22% at curing age of 3 and 90 days, respectively. |
| | [71] | 4 | Improved the strength by 18% at 28 days of curing age. |
| | [72] | 2<br>3 | Improved the strength by 17 and 36% at 7 and 28 days of age, respectively.<br>Improved the strength by 11 and 16% at 7 and 28 days of curing age, respectively. |
| | [73] | 3 | Improved the strength by 50% at a late age of 28 days. |
| | [74] | 2<br>3 | Improved the strength by 43.5 and 58.5% at 7 and 28 days of curing age, respectively.<br>Improved the strength by 82.5 and 48% at 7 and 28 days of curing age, respectively. |
| | [75] | 3 | Improved the strength by 43.4% at 28 days of curing age. |
| | [76] | 2<br>4 | Improved the strength by 22 and 24% at 7 and 28 days of age, respectively.<br>Improved the strength by 26 and 31% at 7 and 28 days of age, respectively. |
| | [77] | 3 | Improved the strength by 18.8% at 28 days of curing age. |
| | [78] | 1 | Improved the strength by 6% at 28 days of age. |
| Nano-titanium | [52] | 5 | Reduction in setting time by 45%. |
| | [79] | 5 | Reduction in flowability by 31.2%. |
| | [80] | 3 | Improved the strength by 15% at 28 days of curing age. |
| | [81] | 1 | Improved the strength by 18% at 28 days of age. |
| | [82] | 1.5 | Improved the strength by 23% at 28 days of curing age. |
| | [83] | 3 | Improved the strength by 61.9% at 28 days of age. |
| | [84] | 2 | Improved the strength by 52.4% at 28 days of age. |
| Nano-zinc | [85] | 5 | Reduction in setting time by 22%. |
| Nanoclay | [86] | 3 | Reduction in flowability by 14.3%. |
| Nano-aluminum | [87] | 2 | Reduction in flowability and setting time by 70% and 55%, respectively. |
| | [29] | 3 | Reduction in setting time by 7%. |
| | [41] | 1 | Improved the strength by 10% at 28 days. |
| | [88] | 3 | Improved the strength by 16.6% and 18.7% at 28 and 90 days of age. |
| | [82] | 1.5 | Improved the strength by 55% at 28 days of age. |
| Nano-iron | [89,90] | 2 | Reduction in workability and setting time by 60% and 55%, respectively.<br>Improved the strength by 15.5% at 28 days of curing age. |
| | [91] | 3 | Improved the strength by 26% at 28 days of curing age. |
| | [92] | 5 | Improved the strength by 57% at 28 days of age. |
| | [82] | 1.5 | Improved the strength by 29% at 28 days of curing age. |
| | [93] | 3 | Improved the strength by 67.3% at 28 days of age. |

**Table 1.** *Cont.*

| NP Type | Ref. | Level (%) | Findings |
|---------|------|-----------|----------|
| CNTs | [94] | 1.2 | Improved the strength by 11.9% at 28 days of age. |
| | [95] | 0.5 | Improved the strength by 32% at 28 days of age. |
| Nanoglass | [58] | 5 | Improved the strength by 17% at 28 days of curing age. |
| Nano-zirconium | [82] | 1.5 | Improved the strength by 20% at 28 days of age. |
| Nano-copper | [70] | 3 | Improved the strength by 51 and 21% at 3 and 90 days of age, respectively. |

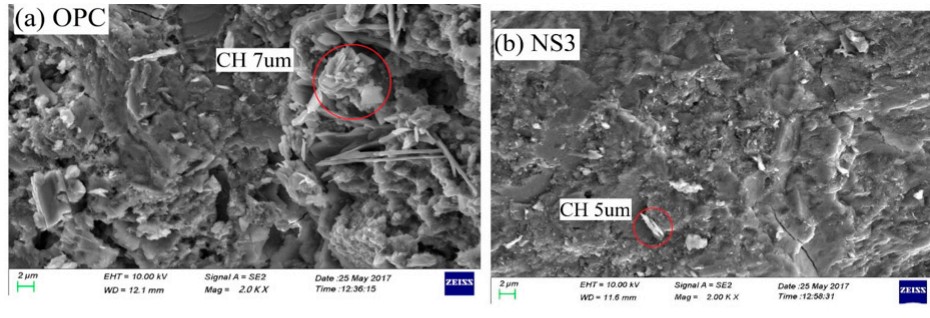

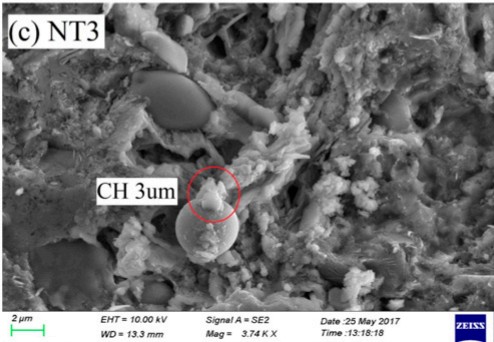

**Figure 4.** Surface morphology of cement (**a**) pure, (**b**) with 3% nanosilica, and (**c**) with 3% nano-titanium [65].

## 3. Sustainability Performance

Chloride exposure that causes the corrosion of the embedded steel remains a major problem for long-term use of concrete systems. The spread of chloride ions in the concrete causes rapid corrosion of the steel reinforcement, generating many cracks in the reinforced steel and eventual splitting of concrete. The chloride ion transport mechanism is a complex system and possibly includes water diffusion, impregnation, and capillary absorption. Consequently, solutions to address this issue have generated much research interest. Many studies have demonstrated that the presence of NPs in the concretes can substantially hamper the chloride ion penetration, thereby protecting the steel structure against oxidation. According to Said et al. [31], the addition of 6% nanosilica into the cement matrix can enhance the durability and reduce the chloride penetration depth by 69.6%. Madandoust et al. [70] reported that the inclusion of nano-additives can reduce the chloride permeability [96]. In comparison to the control specimens, the inclusion of 3% of nano-$SiO_2$, 2% of nano-$Fe_2O_3$, and 4% of nano-CuO was shown to reduce the corresponding chloride permeability by 60, 44, and 44%. Lee et al. [97] examined the chloride ions penetration in mortar incorporated with CNTs (varying contents) and nanosilica (fixed content of 1%). In terms of chloride penetration, the durability of the prepared specimens was found to significantly improve with the inclusion of nanomaterials. The observed improvement in the penetration of chloride ions with the inclusion of CNTs was mainly due to the filling of micropores by the dense gel, bridging the capillary pores in the matrix [98]. The impacts of

zero-dimensional (0D, called nanodots) nanoparticles of $SiO_2$, $Al_2O_3$, $Fe_2O_3$, and $CaCO_3$ incorporation on the chloride ions penetration in mortar were examined. A majority of these studies confirmed that chloride penetration resistance can be improved when the amount of added NPs into the concrete matrix is small. The observed improvement may be due to the NPs' ability to reduce the permeability and porosity of the concrete/mortar matrix that arise from NPs' pore-filling capacity and microcracks refinement [70,99,100].

Another chemical attack due to sulphate can negatively influence the robustness of the reinforced concrete. This attack can be explained as chains of the complex chemical reactions between $SO_4^{2-}$ and cement hydration products [101]. The sulphate ions react with these products as gypsum to produce ettringite, causing the concrete matrix to expand and crack because of the pressure caused by the crystal development [102], swelling [103], volume expansion of solids [104], and topochemical reactions [105]. Expansion and cracks cause concrete to lose strength, thus causing crumbling and weakening of the cementitious mass as time passes [106]. So far, most of the research findings related to the sulphate and chloride ions attack in NP-included concrete composites have shown that the penetration resistance of concrete becomes enhanced with increasing NP content. As aforementioned, this enhancement is mainly due to the capacity of NPs to fill gaps and pores in the dense gel. Accordingly, these NPs function as kernels that refine the micropores and cracks of the concrete matrix, thereby enhancing the resistance against sulphate and chloride ion attacks [107–109].

Currently, an industrial revolution is ongoing in the construction industry wherein various construction materials obtained using the nanoscience and nanotechnology routes are creating sustainable economic development. Many studies have indicated that nanomaterial-based concretes are beneficial for future environmental sustainability because they are energy efficient, can produce clean energy, provide strong and robust performance, have improved resistance against acid attacks, and are eco-friendly with self-cleaning and self-healing attributes [110–112]. Sustainable development is a key issue in the 21st century, wherein nanomaterial-based concretes can make a positive contribution to produce new eco-friendly building products, thus provide a comprehensive understanding of the nano-concretes interaction with the environment. Furthermore, chemistry knowledge and power must be harnessed to ensure that diverse products and technologies for the construction industry based on different nanomaterials are more environmentally friendly and economic than conventional cement-based products. Essentially, construction nanomaterials that are derived from nanotechnology are potentially advantageous due to their strength, durability, and sustainability.

## 4. Carbon NP-Based Cementitious Materials

The most commonly used materials in the infrastructure are cementitious composites [113–115] because of their extreme water-resistance, easy shaping and sizing, low cost, and abundance. Global uses of cementitious composites are over 100% greater than the total of all other construction materials including aluminum, steel, wood, and plastic [115]. In fact, OPC-based composite materials (including concrete and its numerous offshoots) are effective in terms of mechanical performance. These materials typically demonstrate low tensile and flexural strength. Cement composites tend to be quasi-brittle, and consequently, cracks are prevalent due to residual tensile stresses [116,117]. To overcome this weakness on a macro level, diverse types of fibers (glass, steel, and carbon) can be integrated during the mixing process that positively impact the long-term mechanical performance of cement composites, including an improved resistance to crack development during its lifetime. Although they can delay the onset of microcracks, they cannot stop their instigation [118–120]. This is because of the substantial spacing between fibers, implying the free development of microcracks within that space [121,122]. Accordingly, nanofibers and their role in restricting the onset of cracking at the nanoscale has become an interesting research topic.

There are three key benefits to the utilization of various nanomaterials into concrete composites. Firstly, concrete with an excellent CS and thus high-performance concrete can

be produced for a specific purpose. Secondly, a reduced quantity of cement is required in concrete to reach comparable strength levels, thus lowering the construction material cost and adverse impact on environment. Thirdly, construction is faster, as nanomaterials fACI, USAlitate the production of high-strength concrete with more rapid curing [56]. Numerous studies have been conducted to determine the chemical, electrical, mechanical, and thermal properties together with the effectiveness of the nanomaterial reinforcement of concrete composites [123]. Nanoparticles can offer several functions including heterogeneous nuclei for cement pastes (their high reactivity speeds up the cement hydration process), nano-reinforcement, and nano-filling for increasing the density of the microstructure, which results in a decrease in porosity [124].

The Van der Waals interactions in nanomaterials make their functions in cement composite extremely intriguing, wherein the nanomaterials become highly entangled in the matrix due to this force [125]. Figure 5 illustrates a weak interfacial bond between the CNTs and matrix in various regions of the material with numerous cavities. In this view, the effects of nanomaterials' lengths on their dispersion traits in concrete matrix were examined by Yazdanbakhsh et al. [125]. Cement particles are much larger than nanomaterials, causing problems in relation to the size compatibility. This is because nanomaterials are not capable of penetrating the cement grains' area. Consequently, the contents of nanomaterials in several other areas are higher due to clumping, causing weak dispersion. This weak dispersion creates unreinforced areas that facilitate the propagation of cracks, negatively impacting the cement composites' mechanical properties and thus reducing the CS, flexural strength (FS), and splitting tensile strength (STS). Briefly, the dispersion process is highly significant for the cement composites' mechanical properties.

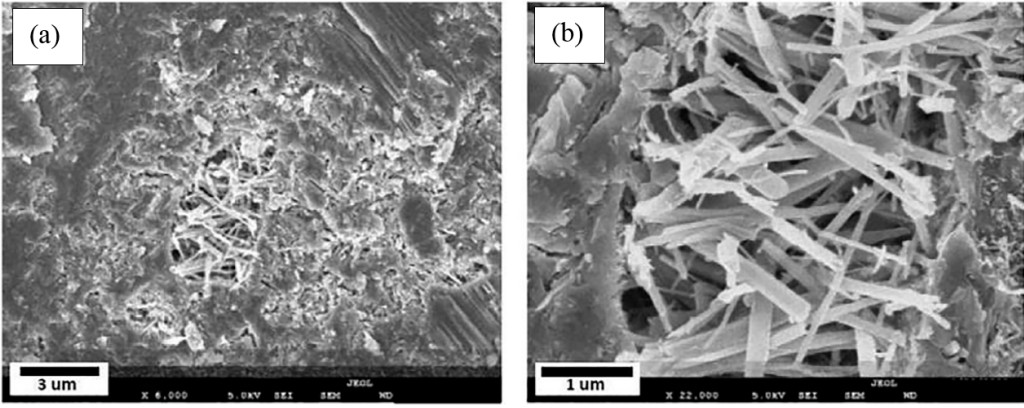

**Figure 5.** Effect of CNT incorporation on cement morphology (**a**) magnification 3 nanometer (**b**) 1 nanometer [113].

## 5. Nanomaterial-Based Geopolymer Concrete

With the ongoing environmental concerns worldwide, sustainable development through the replacement of OPC in the construction sectors have become a key priority [126]. Geopolymer (GP) technology, which was developed by Davidovits in 1970 in France [127], is an expedient replacement for traditional concrete. Historically, geopolymer was used in Ancient Rome to build castles and monuments [128]. GPs belong to the inorganic aluminosilicate polymer family, the members of which are synthesized via the alkaline activation of diverse aluminosilicate materials or other industrial or agricultural by-products enriched in silicon and aluminum (FA, ground blast furnace slag (GBFS), metakaolin (MK), palm oil fuel ash (POFA), and rice husk ash (RHA)) [129]. A polymerization model for the alkali activation of MK was suggested for the creation of zeolites or zeolite precursors from solutions of alkali $Al_2O_3$-$SiO_2$ [130]. Polymerization refers to the chemical reaction between the alkaline activator solution (for instance NaOH and $Na_2SiO_3$) and the source binder material (such as FA and POFA). The outcome is a 3D polymeric chain and ring structure of Si-O-A1-O bonds [131]. Moreover, the aluminosilicate

chains can be present as: (i) poly-(sialate), wherein the Si to Al ratio equals 1.0 (-Al-O-Si-chain); (ii) poly (sialate-siloxo), wherein the Si to Al ratio equals 2.0 (-Al-O-Si-Si- chain); (iii) poly(sialate-disiloxo), wherein the Si to Al ratio equals 3.0 (-Al-O-Si-Si-Si- chain) [132].

Fundamentally, nanotechnology is the capacity to observe and reorganize matter at the atomic and molecular levels within the range of 1–100 nm. In addition, nanotechnology exploits unique traits and phenomenon at length scales comparable to those associated with single atoms and molecules or even their bulk counterparts [110]. In recent years, the topic of nanotechnology has become increasingly popular amongst researchers due to the feasibility of new scientific and practical applications. This has opened up new avenues for integrating various NPs into construction materials in order to boost their properties, producing concrete composites that perform more effectively [133]. Researchers have focused on nanocomposite-based construction materials due to the distinctive physical and chemical characteristics of NPs attributable to their ultrafine particle morphology [134]. NPs of silica [75], $Fe_2O_3$ and $Fe_3O_4$ [135], $Al_2O_3$ [136], $TiO_2$ [137], SWCNTs [138], $CaCO_3$ [139], clay [140], and MWCNTs [141] have frequently been used to improve the performances of conventional cement-based concrete composites. Additionally, various NPs have been integrated with geopolymer matrices in order to improve the their durability, structure, and mechanical traits [142],. The larger surface area to volume ratio of NPs makes them extremely reactive with significantly high reaction rates [143]. Accordingly, the microstructures of geopolymer concretes at the atomic level can be remarkably altered via the inclusion of NPs into the GP matrix. The presence of NPs in the GP matrix can cause substantial enhancement in the morphological and structural properties in both fresh and hardened states without any heat generation [68]. However, some research has utilized nanoparticles in GPs as a functional mixture for non-structural applications including energy storage, self-cleaning, and antibacterial applications.

Earlier, dedicated efforts were made to enhance the various properties of GP composites properties via the insertion of diverse nanomaterials including nanosilica, nano-aluminum, etc. The results in Figure 6 indicate that nanosilica is the most commonly used material in the GP composites. Furthermore, due to the effective pore-filling ability and pozzolanic reactivity of NS, it has also been used in cement and traditional concrete composites [144]. The main component of NS is silicon dioxide ($SiO_2$) available in both crystalline and amorphous states. Typically, NS in its amorphous state was used to develop various types of concretes [145]. NS is comprised of spherical particles or microspheres with an average diameter of 150 nm and a high specific surface area of 150–250 $m^2/g$ can be generated by vaporizing silica (quartz) in an electric arc furnace in the temperature range of 1500–2000 °C. Generally, the sizes of NS particles vary from 5–658 nm for the production of diverse NS-based concrete composites [146]. Five key changes and improvements in concrete occur due to the addition of NPs: (i) polymerization and hydration of the source binder materials, (ii) improvement in the mechanical properties, (iii) densification of the composites microstructures, (iv) lowering of the water permeability, (v) enhancement of the composites' resistance against extreme environmental conditions.

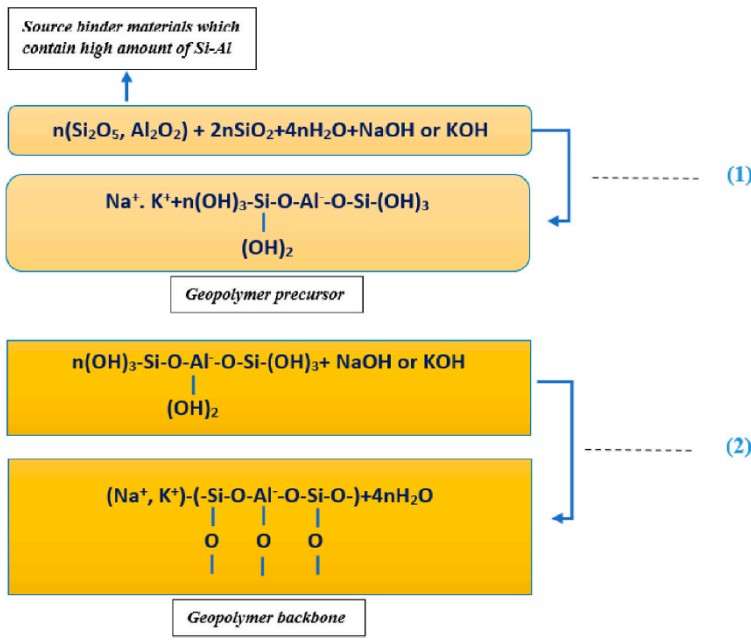

**Figure 6.** Various stages of geopolymerization process [147].

The engineering traits of FA-based GP concretes integrated with nanosilica at various volumes have intensively been studied. The results showed that the mixes devoid of nanosilica achieve higher CS values than the mix integrated with nanosilica. It was found that composite made using 1.5% of nanosilica as an additive has highest strength, which was increased by 11% above the control mix after 28 days of curing at ambient temperature. The observed CS improvement was mainly due to the filling of the nanopores inside the GP concrete by silica NPs, thereby increasing the compactness and density of the matrix. Moreover, the chemical composition of nanosilica with high levels of silica speeds up the GP reactions and strengthens the binder, eventually improving the strength of the specimen. Thus, 1.5% of nanosilica is considered to be the optimal content level for strength enhancement of the concrete composite; any amount beyond this level can cause a decrease in the strength, as many unreacted nanosilica particles in the matrix may couple with the excess nanosilica, causing agglomerations between the nanosilica particles that could otherwise have prevented the dissolution of the silica [148]. This reduced CS of the GP concrete can cause the generation of voids in the matrix [148]. Many studies have confirmed that the strength of GP concrete is enhanced via the integration of nanosilica [149]. Nuaklong et al. [150] added up to 2% of nanosilica into GP concretes and observed an increase in the CS. However, the strength decreased once the nanosilica contents exceeded beyond 2%.

Lincy and Velkennedy [151] determined the cutoff level of NS (about 0.5%) in GP concrete based on the CS improvement results. Ibrahim et al. [152] and Janaki et al. [153] claimed that a 5% threshold exists for the CS improvement in GP concrete due to the addition of NPs, and after this threshold its begins declining. The CS values of concrete at 28 days were increased by 1.5, 13.6, and 1.3% due to the addition of CNTs at 2, 5, and 10%, respectively [153]. Meanwhile, the concrete made with 1, 2.5, 5, and 7.5% of NS showed an increase in the corresponding CS values by 0, 8.2, 23.3, and 19.8% at 28 days of curing age [152]. Kotop et al. [154] observed that a maximum strength of GP mix can be attained with the addition of 2.5% of nanosilica, and the strength was also found to decrease beyond a 2.5% NS content. This decrease was mainly due to strong agglomeration of NS and weak dispersion into the concrete matrix [155]. In addition, in comparison to the control mix, the CS of the GP mix was improved by approximately 81 and 57% at 28 and 60 days curing, respectively, when 0.02% of CNTs was added into the studied mix [154]. This increase in the strength of GP mix was ascribed to the CNT inclusion-mediated bridging of both

micropores and macrocracks in the matrix [156]. The alkaline liquid has an impact on the dispersion of CNTs, as they were the surfactant of sodium hydroxide, enabling them to generate effectively dispersed CNTs and de-bundling within the GP concrete matrix [157]. In addition, Kotop et al. [154] recorded a maximum strength enhancement of 98.7% at 28 days curing age for the GP mix made from 0.01% of CNTs and 2.5% nanosilica.

## 6. Effect of Nanomaterials on Alkali-Activated Binders

Several researchers have studied the effects of nanomaterial addition into the alkali-activated binder in improving the mechanical properties. The observed improvement can be explained using three core aspects: (i) impact of nanomaterial-assisted filling of voids and cracks, (ii) hydration through nucleation sites, (iii) bridging of cracks. The following section offers a comprehensive overview of the latest findings about how the mechanical properties of alkali-activated binders are impacted due to the addition of various nanomaterials such as nanosilica, nano-aluminum, nano-titanium, and CNTs.

Nanosilica can be produced using many approaches including the sol–gel process [158], thermal decomposition technology [159], and vapor-phase deposition [160]. Due to its strong pozzolanic reactivity and pore-filling effect, the use of nanosilica is prevalent in various cementitious mixtures, facilitating enhanced concrete hardening [161]. Nano-$SiO_2$ has very fine spherical textures with an average diameter of 4 nm and a comparatively high surface area to volume ratio. These characteristics are responsible for its high reactivity in geopolymerization reactions [36]. Nanosilica in alkali-activated binders was used to determine its role in accelerating the hydration and polymerization reactions, thus improvement in the mechanical and microstructure properties of binders [162]. It is established that nanosilica has three key impacts on the alkali-activated binders: (i) fast hydration, (ii) refined microstructures, (iii) improved mechanical properties [67].

The most commonly used nanomaterial in the alkali-activated binders is nano-$SiO_2$ [163]. It was acknowledged that addition of 1% of nanosilica with metakaolin GP can cause a substantial increase (about 52%) in the CS at 60 days. However, an increase in the additive to 3% caused a decrease in the CS [164] which was attributed to the weak dispersion of unreacted NPs into the concrete matrix and production of nano-interaction energy. The optimal level of nanosilica addition in coal FA/slag binder-based GP was shown to be approximately 2% (Figure 7a). In addition, it was found that the inclusion of nanosilica can cause a minor delay in the hydration and additional calcium–alumina–silicate–hydrate gel formation with a decreased porosity of the alkali-activated GP [165]. It was demonstrated that nanosilica addition can improve the extent of geopolymerization, ultimately generating a denser geopolymer paste [148]. Figure 7b shows the impact of nanosilica on the CS of FA/GBFS-based alkali-activated specimens. The integration of diverse levels of nanosilica (1–8%) with GPs was also examined, and the results indicated an increase in the CS up to 3% of nanosilica addition. However, beyond 3% nanosilica addition, the mechanical properties were significantly deteriorated due to the nano-agglomeration and declination of geopolymeric gel formation [166].

The wet mix method to add nanosilica was found to be more successful for enhancing the microstructure and mechanical traits of GPs [167]. The widely accepted theory behind the observed improvement was credited to the geopolymerization and stimulation of denser Na Ca-(A)-S-H gel development. It was established that a nanosilica-mediated pozzolanic effect was not only the reason for the alkali-activated binders' strength improvement, but the filler effect of NPs was also responsible for the improvement in the cohesion of the microstructures and strengthening of the aggregate–paste bonding [162]. Furthermore, the nanosilica provided more amorphous silicate to instigate the geopolymerization process, which is the core sources of strength in GP mixes. All of these are surplus in the theory of the nucleation impact of NPs in the creation of aluminosilicate gels [68].

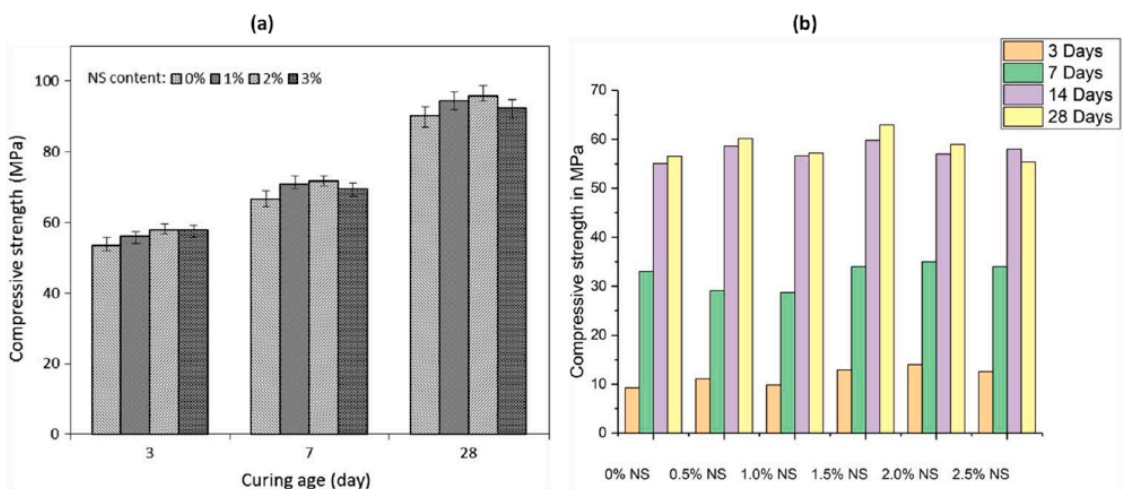

**Figure 7.** Influence nanosilica content on CS of (**a**) GBFS-FA [167] and (**b**) GBFS specimen [148].

It was suggested that nano-alumina can be a suitable additive for cement and concrete because of its thermal stability, high surface area, and unique mechanical characteristics [168]. Nano-$Al_2O_3$ was used in cement-based materials because of its advantages in refining the microstructures, pore compositions, and the ITZ [169]. The influence of nano-$Al_2O_3$ on the mechanical characteristics improvement was mainly due to its physical (filler) effect, resulting in an accelerated early-stage hydration [170]. Some researchers have demonstrated that nano-$Al_2O_3$ has no or only minimal positive impact on the strength properties of concrete [171], which is likely due to their weak dispersion in the cement-based matrix. To verify this claim, the impact of nano-alumina on the geopolymers has carefully been examined [172], and the results revealed that the use of nano-$Al_2O_3$ as an additive cause a decrease in the initial and final setting time required for FA class C' binders wherein an insignificant improvement in the strength level was noted [35]. Phoo-ngernkham et al. [29] found that the inclusion of nano-$Al_2O_3$ into high-calcium FA at 1, 2, and 3% can improve the compressive and flexural strength of the room-temperature-cured specimens. The obtained improvement in the strength properties was ascribed to the generation of extra geopolymerization and hydration products [29]. Incorporation of nano-$Al_2O_3$ at 2% as an additive into GBFS-based alkali-activated mortars was found to enhance the strength performance (Figure 8) at all stages up to four months [173]. A correlation was identified between the contribution made by the nano-$Al_2O_3$ in the aluminosilicate formation and improvement in the CS of the binder. In contrast, use of nano-$Al_2O_3$ additive more than 3% was found to reduce the CS at 7, 28, and 90 days of curing. This drop in the CS was mainly due to the inadequate dispersion of NPs into the GP matrix and formation of weak zones within the GP pastes.

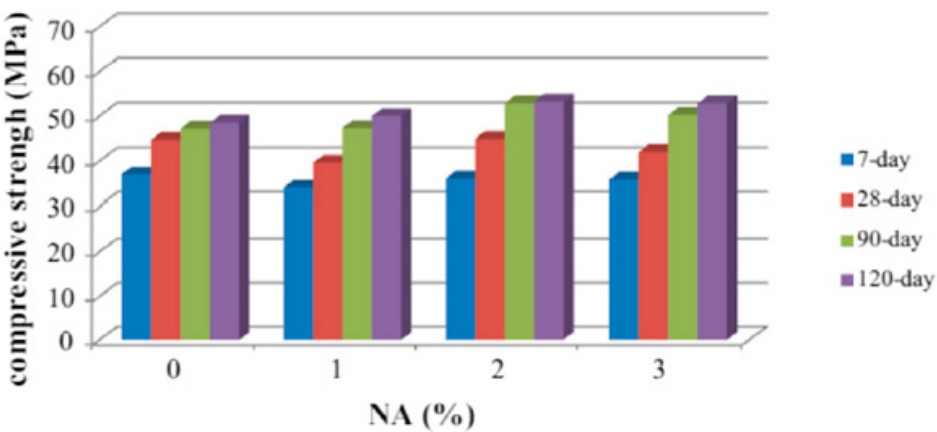

**Figure 8.** CS of alkali-activated slag without and with nano-$Al_2O_3$ inclusion [173].

The effects of nano-titanium dioxide inclusion (fine spherical or ellipsoidal powder-based materials of diameters below 100 nm) on the CS properties of GP mortars have been examined [174]. Nano-TiO$_2$ (a broad band gap semiconductor) has become an increasingly popular additive into cementitious materials for improving the strength and durability properties because these NPs can offer a distinctive aesthetic appearance to the architectural constructions. In addition, TiO$_2$ can decompose a vast array of organic and inorganic air pollutants under ultraviolet light and humidity, thus generating cleaner air and enhanced wellbeing for people and other living creatures. Another advantage of TiO$_2$-based cement/concrete is related to its white color, usually having broad architectural appeal [175]. Many state-of-the-art reports showed that the integration of titanium into the construction materials can generate cleaner air and self-cleaning properties useful for the eradication of detrimental bacteria and germs [176]. Moreover, uses of nano-titanium as an additive into the concrete mixtures can assist in the acceleration of the cement hydration process, thus increasing the durability of the concrete due to a significant reduction in water permeability [177]. In addition, TiO$_2$ is a celebrated photo-catalytic agent that offers strong photodecomposition via light absorption useful for diverse practical applications. The 'self-cleaning' nature of TiO$_2$ can be exploited by mixing with cement wherein such traits can be harnessed on a wide scale to improve the air quality in urban areas.

Recent study has shown that by substituting FA with 1 and 5% of nano-TiO$_2$ in the GP concrete, the CS can appreciably be improved [178]. A comparison of the CS for FA-based concrete (Figure 9) with and without nano-titanium additive at 7 and 28 days of curing showed an increase in the CS with an increase in nano-titanium level having a maximum improvement above 50%. Maiti et al. [179] investigated the impact of nano-titanium particles inclusion at various sizes (30, 50, and 100 nm) on the FA-based GP made at ambient temperature (Figure 10). At 28 days of curing, the GP mix prepared with 30 nm of nano-TiO$_2$ (5%) showed a maximum STS of approximately 7 MPa (Figure 11). Duan et al. [79] noted that the strength improvement was more faster during the early stages (up to 28 days) than the later stages (up to 90 days). This revelation was attributed to the NP-induced geopolymerization process and densification of the microstructures of GP mix.

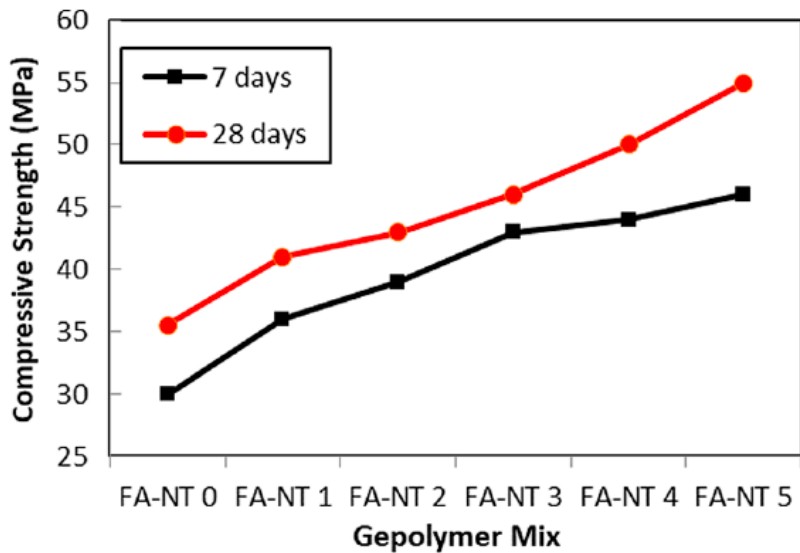

**Figure 9.** Strength development of FA-based mixes enclosing various levels of nano-TiO$_2$ [178].

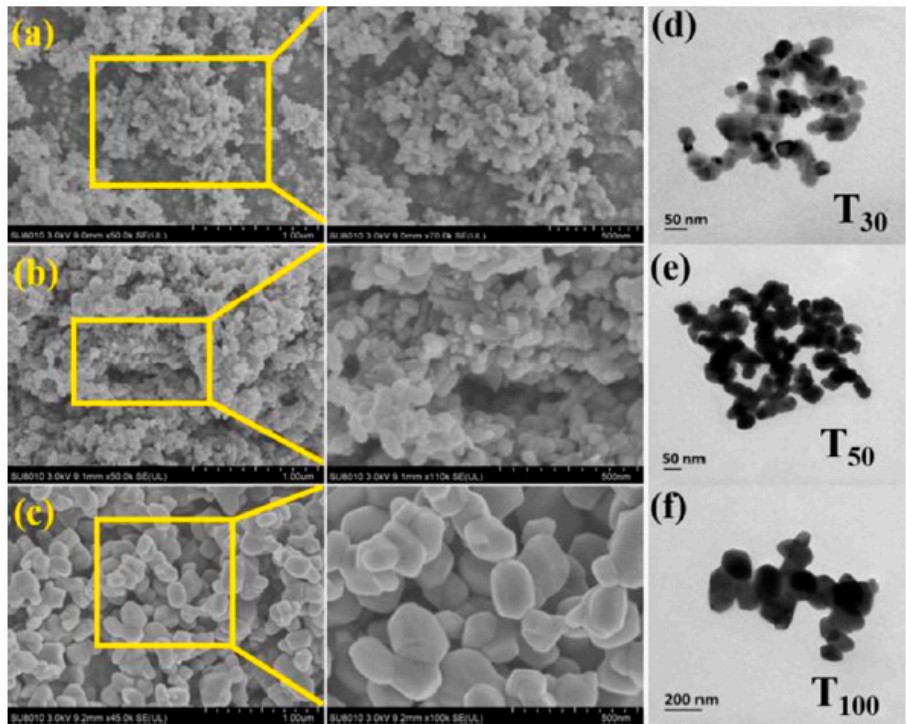

**Figure 10.** Surface morphology of alkali-activated specimens containing various level of nano-titanium (**a**,**d**) T30 (**b**,**e**) T50 and (**c**,**f**) T100 [179].

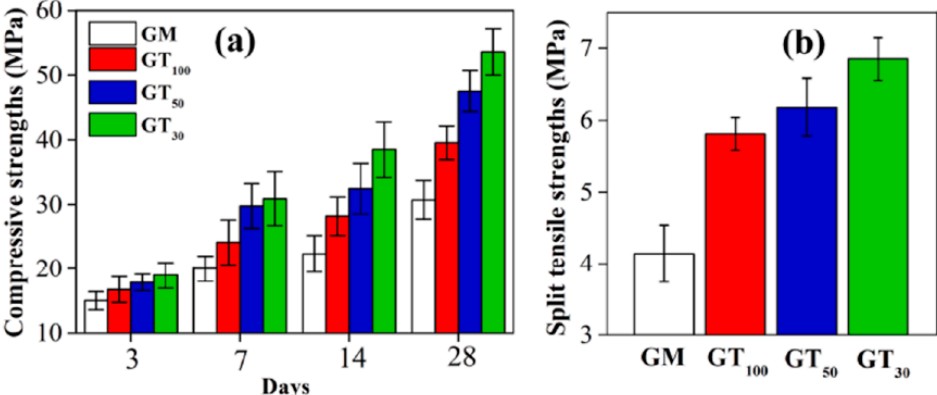

**Figure 11.** (**a**) CS and (**b**) STS development of alkali-activated specimens made with various nano-titanium contents [179].

## 7. Conclusions

This critical overview on NP-based concrete composites enabled the following conclusions:

i.  Inclusion of NPs into the mix can reduce its flowability as well as both initial and final setting times. Significant improvement in the hydration rate with the inclusion of NPs can lead to the reduction of the setting times of the concrete.

ii. Mixes prepared with nanomaterials show excellent improvement in the strength performance. Inclusion NPs in the cement/cement-free binders can cause significant development in the dense gels and dense surface morphology.

iii. Nanosilica is widely used in concrete technology and recommended for several applications. It can improve the early strength, reduce the porosity, and enhance the corrosion resistance of modified concrete composites. An optimum dosage of nano-$SiO_2$ between 1 to 6% was suggested depending on the nature of concrete and

mix design which could enhance the early and late strength up to 28% and 10%, respectively.

iv. The durability traits (porosity and chloride penetration) of NP-modified concrete composites can be enhanced via the inclusion the NPs into the concrete matrix.

v. Incorporation of CNTs in the OPC composite can improve their engineering traits more compared to other nanomaterials.

vi. Inclusion of nanosilica in cement-free binder was found to promote geopolymerization reactions, shortening the setting time, and enhancing the durability and mechanical properties of the resultant geopolymer binders.

vii. Increase in the NPs content more than the optimum percentage can negatively affect mechanical and durability characteristics of concrete due to the difficulty of uniform dispersion and formation of weak spots within the binder matrix.

viii. In short, careful utilization of various potential nanomaterials as additives can be a novel strategy to enhance the microstructures of cementitious components that can eventually lead to sustainable developments in construction sectors worldwide.

**Funding:** This research received no external funding.

**Data Availability Statement:** Not applicable.

**Acknowledgments:** The authors thank the National University of Singapore for their support and cooperation in conducting this research.

**Conflicts of Interest:** The authors declare no conflict of interest.

## Abbreviations

| | |
|---|---|
| NPs | Nanoparticles |
| $CO_2$ | Carbon dioxide |
| CNTs | Carbon nanotubes |
| C-S-H | Calcium silicate hydrates |
| CS | Compressive strength |
| SF | Silica fume |
| NS | Nanosilica |
| ITZ | Interfacial transition zone |
| FS | Flexural strength |
| STS | Splitting tensile strength |
| GP | Geopolymers |
| SEM | Scanning electron microscopy |
| FA | Fly ash |
| GBFS | Ground blast furnace slag |
| OPC | Ordinary Portland cement |

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
