# Peer review of "A Review on Concrete Composites Modified with Nanoparticles"

_jcs, doi:10.3390/jcs7020067_

Round 1

Reviewer 1 Report

The title of the manuscript is grammatically incorrect and therefore unclear to the potential reader. Instead of "A Review on Nanoparticles Modified of Concrete Composites" it should be, for example, "A Review on Concrete Composites Modified with Nanoparticles".

Please indicate whether this review brings anything new compared to other reviews cited in the manuscript, for example [25], [44] and [54]? It's hard for me to find arguments for this statement.

The abstract is too general, especially in the first half, and does not correspond to the content of the manuscript. It should contain the most important conclusions and a summary resulting from the review of many papers so as to encourage the reader to read the article.

The introduction is also off topic, it focuses on issues related to CO2 reduction and important problems are treated superficially.

Substantive error (lines 262-263): " ... can substantially hamper the carbonation (the negative impacts of chlorine ions),", this is a compromising error for the author: carbonation is not the negative impact of chloride ions (on what?) it is completely another phenomenon.

There is a lack of order in the use of abbreviations. After entering an abbreviation in the text, it should be used later in the text. The impression is as if the entire text was composed of at least two separate texts, as well-known abbreviations are introduced anew (this applies in particular to the chapter on geopolymer concrete). This is confirmed by the fact that the issue of CO2 reduction is discussed again in the beginning (from line 364) of the chapter mentioned at the beginning of the manuscript.

There are editorial and other errors:

1) lines 64 and 65

2) line 187 Cu2O3 ? (so far no such compound has been isolated)

3) line 416 (dot unnecessary)

4) line 469 (the sentence is not clear)

5) line 540 (what does ITZ mean?)

6) line 582 (is nanotitanium the same as nano-TiO2?, if so, please do not use the term nanotitanium, it is misleading)

7) deficiencies or errors in the bibliography: for example, lines: 643, 674, 915.

Author Response

Reviewer' comments are highly appreciated.

Attached please find the author response to reviewer' comments. 

Author Response

Reviewer' comments are greatly appreciated.

Attached please find our response to reviewer' comments. 

Round 2

Reviewer 1 Report

The author acted on some of the comments and revised the original manuscript in the suggested direction. This applies to comments #1-1, #1-3, and #1-4. However, in the other answers, they wrote that they took into account the comments in the text and did not take them into account, and at the same time, they give false line numbers of the changes made. Comment #1-5 on carbonation is important. It is nonsense that this is the negative effect of chloride (authors write “chlorine” – error)  ions. Carbonation is a reaction of hydroxides contained in concrete with CO2 leading to a decrease in the pH of the concrete cover. So it's something completely different.

In comment #1-7, the errors remained uncorrected, now on lines 52-53. In comment #1-9 I wrote that you need to remove the dot in the middle of the sentence - the author removed the entire sentence, why?

Comment #1-13 part of the bibliography the authors removed, the rest they did not correct, e.g. now point 119 (line 865) - a false page is given, now point 12 (line 626) needs to be removed "In Proceedings of the", Micro is the name of the journal.

The entire manuscript must be carefully reviewed and corrected.

Author Response

Reviewer' comments are highly appreciated and author response as attached. 

Reviewer 2 Report

The paper is considerably improved, mistakes have been removed, and the manuscript has been rewritten in several points.

I have only one remark:

In line 90, instead of "cement industry", the correct is to write "concrete production".

Author Response

(The authors gave the same response as above.)

Round 3

Reviewer 1 Report

This time the author has corrected the mistakes I pointed out earlier. I believe the manuscript may be published.

Reviewer 2 Report

Dear authors

In the manuscript I received I see  in line 90 "cement production" instead the correct "concrete production"

Please follow correctly my remark. 

In line 621 I see also an additional error:

GBFS: Ground blast furnace slag. The correct is "granulated blast furnace slag"